# A novel application tunnel in combination with medical training reduces stress induced by frequent intraperitoneal injections and blood draws in mice

Anne Schlutt[1,2], Katarina Riesner[2,3], Merle Kochan[4], Ann-Kathrin Meß[4], Denise Jahn[2,5], Isabella Lurje[1], Wiebke Werner[1], Felix Heymann[1], Olivia Kershaw[6], Frank Tacke[1], Bernhard Hiebl[7], Linda Hammerich[1,2]*

**1** Department of Hepatology and Gastroenterology, Charité - Universitätsmedizin Berlin, Campus Virchow-Klinikum and Campus Charité Mitte, Berlin, Germany, **2** Taskforce Refinement, Charité - Universitätsmedizin Berlin, Berlin, Germany, **3** Department of Hematology, Oncology and Tumorimmunology, Charité - Universitätsmedizin Berlin, Berlin, Germany, **4** Julius Wolff Institute, Charité - Universitätsmedizin Berlin and Berlin Institute of Health at Charité, Berlin, Germany, **5** Department of Oral and Maxillofacial Surgery, Charité - Universitätsmedizin Berlin, Berlin, Germany, **6** Institute of Veterinary Pathology, Freie Universität Berlin, Berlin, Germany, **7** Institute of Animal Hygiene, Animal Welfare and Farm Animal Behaviour (ITTN), University of Veterinary Medicine Hannover, Hannover, Germany

* Linda.hammerich@charite.de

## Abstract

Handling-induced stress represents a major burden on laboratory mice and is heavily influenced by the handling technique. As such, tail-handling induces much higher stress than cup- or tunnel-handling, which is further aggravated by interventions, e.g., injections, which require even harsher fixation methods. Previous studies demonstrated that habituation protocols can improve animal welfare during handling and interventions. Here, we developed a medical training program and an alternative fixation method using an application tunnel in the context of a liver cancer model, where frequent intraperitoneal injections in male C57Bl/6 mice over many weeks are needed to induce tumor development. The training regimen consisted of 5 sessions over 2 weeks, which gradually introduced the animals to being touched by handlers and restrained for procedures. The training program was completed once before the start of any interventions, additional training sessions were performed biweekly during the entire course of the experiments. The animals were randomized to receive injections either in the novel application tunnel or using conventional fixation. Training effect was continuously monitored by measuring the latency to interact with the experimenter, the surface body temperature and by activity tracking. The latency to interact rapidly decreased during initial training sessions in both groups, and this effect was sustained throughout the course of treatment. Body temperature did not change over the course of medical training and seemed to be more influenced by environmental influences. Activity tracking demonstrated that mice injected inside the tunnel were more active and returned to their normal activity levels after injections

**Data availability statement:** This study protocol has been preregistered on animals-tudyregistry.org and is accessible under this DOI: 10.17590/asr.0000307. The data that support the findings of this study, technical diagrams and 3D-printing files for manufacturing the tunnel as well as a video demonstrating the injection procedure have been deposited on the Open Science Framework platform (OFS. io) and can be accessed under this link: https://osf.io/vsjnb.

**Funding:** This work was funded by Charité 3R – Replace, Reduce, Refine (https://charite3r. charite.de/), grant awarded to LH. The funder was not involved in study design, data collection, analysis, decision to publish or preparation of the manuscript.

**Competing interests:** A.S. and L.H. own a utility model for the injection tunnel in Germany (Registration No. 20 2024 100 448). All other authors declare no conflicts of interest.

faster than conventionally restrained mice. Those mice also showed less signs of defecation and urination during interventions. Furthermore, the tunnel had a positive influence on the well-being of mice during blood draws demonstrated by reduced signs of pain and faster willingness to interact with the handler after the procedure. In conclusion, habituation of the mice to the interventions with medical training and the improved handling procedure during ip injections and blood draws durably reduces stress levels and improves welfare of mice.

## Introduction

Laboratory animals are still indispensable for medical research to understand the function of genes, the mechanisms of diseases and the effectiveness and toxicities of drugs and chemicals. This is particularly relevant for cancer research, as many hallmarks of cancer can only be fully recapitulated in models using living organisms [1]. The house mouse (Mus musculus) is the most frequently used animal species for biomedical research in the European Union; about 1 million mice were used in 2024 for scientific purposes in Germany alone [2]. Directive 2010/63/EU [3] adapted by the European Parliament and the Council of the European Union aims to standardize and harmonize national legal provisions on animal welfare in research. Member States are encouraged to promote research in the field of the 3Rs, which are embedded in article 4 of the Directive [3]. 3R stands for *"Replacement"*, *"Reduction"* and *"Refinement"* as defined by William Russell und Rex Burch´s *„The Principles of Human Experimental Technique"* in 1959 [4]. Nowadays, replacement of animal studies with alternative strategies is often viewed as the ultimate goal. However, if replacement by animal-free methods is not possible, refinement methods can be used to improve animal welfare in basic and translational medical research.

Identifying and implementing the most appropriate refinement methods is always dependent on the individual experimental design. It may include enhanced cage enrichment, proven stress-reducing handling methods like cup or tunnel handling or a habituation of the animals to the interventions during experiments in form of a training program. Handling is the procedure most frequently experienced by laboratory animals (during regular husbandry as well as experimental interventions). Especially handling by less-experienced personnel can lead to repeated attempts during handling sessions with an increased risk of being bitten, increased experience of stress and a higher risk of injuries to both the animal and the handler due to wrong restraining or mis-application (needlestick injuries). Ultimately, handling-induced stress may influence the results of the respective research question [5,6]. Therefore, it appears promising that the implementation of a medical training could be a useful tool to improve animal well-being in a laboratory context since laboratory routine itself is stressful to mice [5]. It has been shown that habituation to the method of handling can lower long-term stress and anxiety, as it helps the animals to associate handling with neutral or positive experiences and better tolerate brief unpleasant procedures [6,7]. Habituation involves repeatedly exposing animals to a non-threatening stimulus

such as simple, repetitive handling [8]. Currently available protocols usually employ training programs that last 3–5 days and sometimes include cognitive enrichment methods such as clicker training as well [6,9,10]. Handling can also be trained for specific procedures, such as injections or oral gavage [11], and strategies have been described to refine the immobilization method for these procedures. These include the 3-finger method, which creates less pressure on the throat area than traditional scruffing methods, and a technique without full restraint where mice are allowed to grasp a wire grid while right hind limb and tail are secured with one hand to allow access to the lower abdomen for injection [12,13]. However, the former still requires the mice to be picked up and turned around into an unnatural and therefore fear-inducing position, while the latter provides no protection from being bitten by stressed animals and might limit the amount of time to administer injections, especially in more aggressive strains. It is important to emphasize that the type and frequency of handling depends on the particular experimental setup and any training program must be well thought out to fit the need. Because experimental procedures vary widely, only very limited universal training guidelines exist.

Here, we describe the implementation of a medical training program and a novel application tunnel for injections and blood draws in the context of a mouse model for hepatocellular carcinoma (HCC), which requires repeated intraperitoneal (ip) injections over many weeks. Regarding our training program, the different steps can be adapted depending on the experimental design. In order to determine the training effect, we evaluated several non-invasive stress parameters, including the latency to interact with the experimenter, surface body temperature, corticosterone accumulation in fur as well as the general behavior of the mice. The latency to interact is an easily measurable parameter that can provide immediate information about the stress level and well-being of mice [14], which are flight animals that hide in dangerous situations. The more comfortable a mouse feels, the more voluntary interaction with humans can be observed [14,15]. Another common method to detect acute stress or illness in rodents is measuring the body temperature [16]. Since intrarectal measurements can cause additional stress [17], we chose to measure the surface temperature with a contactless infrared thermometer [18]. We also measured corticosterone accumulation in fur as an end-point parameter, as it has been shown that corticosterone increases in serum and fur not only due to heightened activity or arousal [19] but also as a response to stress [20,21], which can be influenced by the handling method [22,23]. Finally, we evaluated the general behavior of the mice after injections by assessing activity patterns and defecation/urination. After blood draws, the mouse grimace scale (MGS) was applied as a readout for pain.

## Materials and methods

### Animals and housing

C57Bl/6J and C57Bl/6N wildtype mice (age range 14 days to 26 weeks) were bred in-house in the experimental area of the Research Institutes for Experimental Medicine (FEM) of Charité – Universitätsmedizin Berlin. Therefore, no acclimatization period to the experimental area was necessary. Mice were kept in groups in eurostandard type-III-cages made from polycarbonate (425 x 266 x 185 mm, base area 800 cm2, Tecniplast Germany, Hohenpeißenberg, Germany) at a relatively constant temperature (range 20–24°C) and humidity (range 30–70%). Animals had access to food (1324, Altromin Spezialfutter GmBH, Lage, Germany; contains 55% carbohydrates, 19,2% protein, 12% crude ash/fiber, 4,1% fat, moisture content 11,3%) and water (untreated tap water provided in water bottles) *ad libitum* and cages were changed once per week. All cages were enriched with spruce wood granulate (FS 14, Safe-lab, Essingen, Germany), cotton pads (14010, Plexx B.V., Elst, The Netherlands), paper nesting material (Sizzle-Nest, Datesand, Stockport, UK), 2 aspen wood gnawing blocks (H9234-NGS, 40 mm x 16 mm x 10 mm, Ssniff Spezialdiäten GmbH, Soest, Germany), 1 nesting igloo made from red polycarbonate (Mouse House, Tecniplast Germany, Hohenpeißenberg, Germany), 1 opaque grey tunnel for hiding and 1 clear acrylic tunnel (50 x 3 x 100 mm, Datesand Group, Stockport, UK), which was also used for contact-free transfer of the mice between cages during the weekly cage changes. Mice were maintained on a 12/12 h light/dark cycle (6am-6 pm) and experiments were conducted between 4 pm and 7 pm when the animals are more active than in the morning hours. All animal experiments were reviewed and approved by the competent local authority (Landesamt für Gesundheit

und Soziales Berlin (LAGeSo); approval numbers G 0310/19, G 0112/23, G 0131/23, G 0037/21) and were conducted in accordance with the ARRIVE 2.0 guidelines.

## Hepatocellular carcinoma induction

HCC induction was performed in male mice only, because HCC is much more prevalent in men than women and accordingly, early exposure to the DNA-alkylating agent diethyl nitrosamine (DEN) leads to HCC development in almost 100% of male mice, but only in 30% of female mice. This is thought to be due to estrogen [24,25]. HCC was induced based on established protocols [26–28]: male C57Bl/6J mice received a single ip injection of 25 mg/kg DEN at the age of 14 days followed by repetitive (twice weekly) injections of carbon tetrachloride (CCl$_4$; 0.5 ml/kg diluted in corn oil) to induce fibrosis or feeding a Western-style diet (WD, E15723-34, Ssniff Spezialdäten GmBH, Soest, Germany) containing 21% butter fat, 1.25% cholesterol, 17% protein, 47% carbohydrates and 9% crude fibre/ash to induce steatosis from the age of 8 weeks (Fig 1). All male mice born in our colony received DEN injections and were subsequently randomized to receive CCl$_4$ or WD when they were 8 weeks old. The combination of DEN and CCl$_4$ resulted in tumor development within 10 weeks (i.e., at 18 weeks of age), the combination of DEN with WD resulted in tumor development within 16 weeks (i.e., at 24 weeks of age). Tumor-bearing mice were again randomized to be treated with an immunotherapy regimen (either a dendritic cell growth factor, fms-like tyrosine kinase 3 ligand (Flt3L), an immune checkpoint inhibitor, or both) administered via ip injection. Animals were monitored for welfare and health status via daily visual health checks. Efforts to minimize suffering and distress are described under medical training and injection procedures. Death by euthanasia was not a planned experimental endpoint for this study as evaluation of the welfare methods described here concluded at a maximum age of

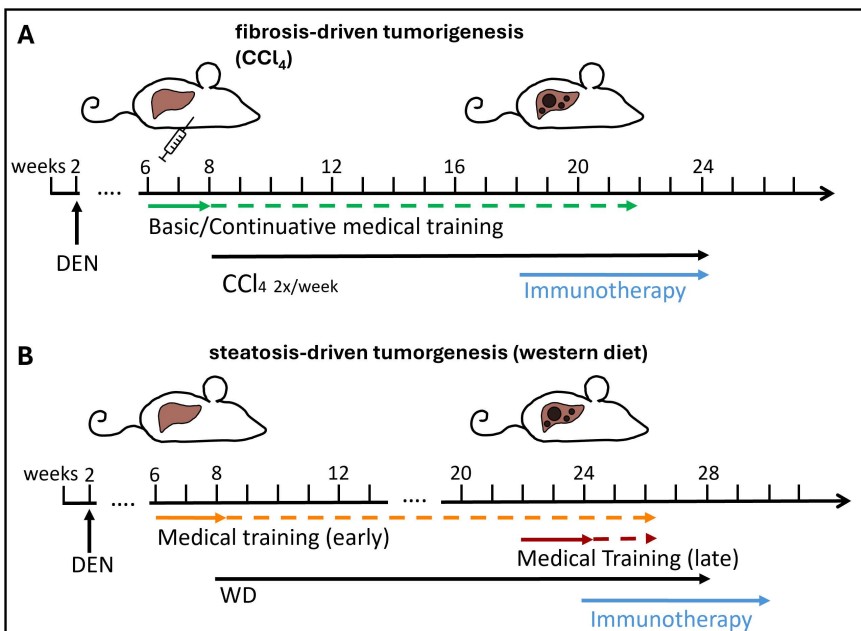

**Fig 1. Mouse models.** To induce tumorigenesis, mice are injected with diethyl nitrosamine (DEN) at the age of 14 days and then subjected to carbon tetrachloride (CCl$_4$) injections to induce fibrosis or western diet (WD) feeding to induce steatosis. (A) fibrosis model: Basic medical training program starts at 6 weeks of age, CCl$_4$ injections start at 8 weeks of age. Training is continued (2 sessions every other week) until immunotherapy treatments commence at 18 weeks after tumors have developed. (B) steatotis model: Basic medical training program starts either at 6 weeks of age („early"; 2 weeks before WD feeding starts at 8 weeks) and is continued (2 sessions every other week) until immunotherapy treatments commence at 24 weeks or starts at 22 weeks („late"; 2 weeks before immunotherapy treatments).

26 weeks (see below) after which the mice continued to be monitored for tumor growth as part of other ongoing studies. To ensure that spontaneous death due to disease-related events could be prevented, humane endpoints and termination criteria were defined in case animals should show early signs of suffering or distress. These included: indicators of lasting pain (more than 24 hours after analgesia), severely impaired overall condition (weight loss >20%, body condition score of 2 or lower, palpable tumors >10 mm, centralized circulation, lack of reaction to stimuli), or largest tumor diameter (of a single nodule) of 15 mm as measured via non-invasive magnetic resonance imaging. Animals were monitored for these endpoints throughout the whole course of the study; none of the animals reached any humane endpoint until conclusion of the study.

## Experimental groups

Following the 3R principle of reduction, this welfare study was conducted exclusively on animals already undergoing HCC induction in other ongoing projects. All animals undergoing HCC induction during the course of this study were included. Within the $CCl_4$ model, mice were randomized to receive medical training and/or ip injections in the novel application tunnel (S1 Table, groups 2–5). Within the WD model, mice were randomized to receive medical training either early (at 6 weeks, i.e., 2 weeks before beginning of WD feeding) or late (at 20 weeks, i.e., 2 weeks before ip injections begin) (S1 Table, groups 6–8). As a baseline control group for biochemical measurements, 22-weeks old healthy mice without any interventions (no disease induction) were included as well (S1 Table, group 1). Because animals entered the experiments at 2 weeks (=before weaning), animal numbers differed between cages depending on the number of male pups per litter. In general, mice in the same experimental groups were housed together. Therefore, we treated cage as the experimental unit, unless otherwise stated. Blinding of the experimenters during training and sample/data collection was not possible, because all animals had to be trained repeatedly and many measurements were taken during or immediately after training. This required the experimenters to be aware of the group allocation of the animals at any given time. However, experimenters were blinded to sample allocation during end-point sample analysis whenever possible (e.g., corticosterone measurements, pathological examination).

## Medical training program

All animals undergoing HCC induction were included in the medical training program and analyzed according to their group assignment (no exclusions of animals occurred). The training program included a basic training consisting of 5 training sessions within 2 weeks (S2 Table and Fig 2) and was started at the age of 6 weeks, i.e., 2 weeks before the beginning of $CCl_4$ injections or WD feeding to induce liver cancer (Fig 1). The amount of training sessions was chosen based on previous studies demonstrating training success after completing 5 training sessions with mice [29]. At the end of every training session mice were given treats (Kellogg's fruit loops®) in the home cage and training sessions were 2–3 days apart to allow time for recovery. The same Vetbed® was used for all mice in one cage (but not shared between cages) and reused over several training days. Vetbeds were cleaned after contamination with urine/feces. A detailed description of the training sessions included considerations for handler behavior, criteria of success and how to respond to animal behavior can be found in S2 Table. In general, handlers performing the training should stay calm, avoid fast movement and not put any pressure on the animals (e.g., corner them) as this can lead to negative associations.

## Conventional Handling (scruffing) for injections

Mice were immobilized on a cage grid by securing the tail and allowing the mouse to grasp the cage grid with its forepaws. Then, the lower abdomen and extremities were gently pushed down with the hand holding the tail while pulling back (scruffing) the fur and skin in the neck area between thumb and forefinger of the other hand. Mice were turned around to expose the abdomen and the tail fixated between the palm and the little finger of the same hand holding on to the neck (Fig 2E), leaving the other hand free for injections [13].

 

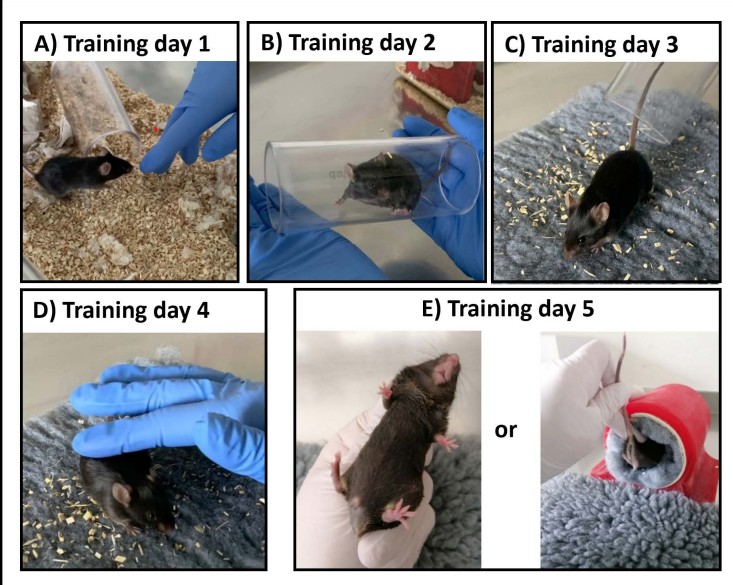

**Fig 2. Medical training program. (A)** Day 1: Gloved hand is reached inside the cage, handler waits for voluntary interaction. **(B)** Day 2: mice are directed into acrylic handling tunnels and held for about 30 seconds before being released back into the cage. **(C)** Day 3: mice are transferred onto a Vetbed outside of the cage using the acrylic handling tunnel. Handler tries to touch the animals gently for about 30 seconds, before transferring them back. **(D)** Day 4: same as day 3, with a focus on touching neck, back, tail area for about 30 seconds. **(E)** Day 5: fixation method for injections is performed once without injection.

## Application tunnel for intraperitoneal injections

The application tunnel was 3D-printed in house, details on the design and manufacturing can be found in supplementary methods and S1 Fig. The size of the tunnel was dependent on the age of mice; for mice under 10 weeks a tunnel with an inner diameter of 7 cm, for older mice of 8 cm was used. The length of the tunnel was 10 cm. A rolled-up piece of a washable and autoclavable Vetbed, which the mice were introduced to during the medical training, was placed inside the tunnel. Since studies have shown that mice feel safer on groundings which have similar color to their fur [30], a grey Vetbed was used (Fig 3A). The same Vetbed was used for all mice housed together in one cage, but changed between cages. After each day of injection both the Vetbed and tunnel were cleaned.

## Injections inside the tunnel

Fig 3 shows the procedure of ip injection using the tunnel (a link to a video demonstrating the injection procedure can be found in supplementary methods). Mice were removed from their home cages using the acrylic tunnel already known from the medical training and placed on a Vetbed outside of the cage. In our experience, the mice often enter the application tunnel voluntarily and instinctively as it offers a place to hide. If not, mice were gently directed to enter the application tunnel (Fig 3B) The handler carefully held the tail's base between forefinger and thumb to prevent that the mouse is walking too deep inside. By applying soft pressure on the sacral area with the middle finger of the same hand, the mouse is immobilized (Fig 3C). This fixation allows a good view on the caudal abdominal region where ip injection is performed (Fig 3D). After completing the injection, the injection tunnel was placed inside the home cage to allow mice to return to their home cage without additional handling.

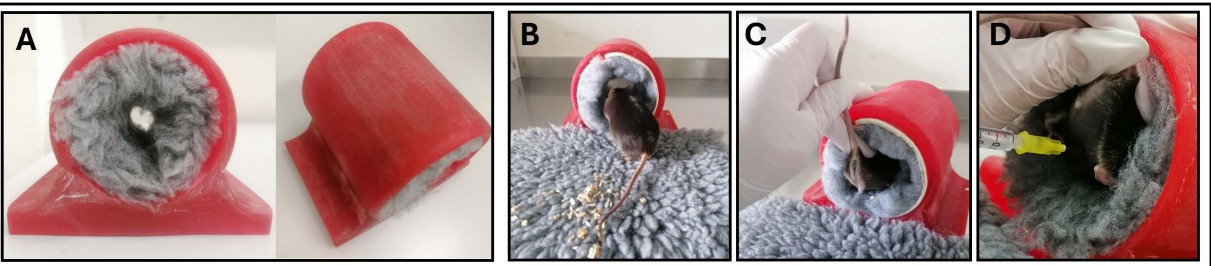

**Fig 3. Application tunnel. (A)** 3D-printed tunnel with Vetbed. **(B)** Mouse entering tunnel. **(C)** Soft pressure is applied on the sacral area and tail fixed between forefinger and thumb. **(D)** View on caudal abdominal region for ip injection and location of injection (syringe).

### Latency to interact with the experimenter

For the purpose of this study, we defined the latency to interact as the amount of time required for a mouse to voluntarily get into contact with the handler after completing a training session. To measure this, mice were placed back into their home cage after each training session or after blood draws and one gloved hand of the handler was reached inside the cage but did not move. A timer was used to measure the time it took for the animal to voluntarily interact with the gloved hand again (e.g., sniff). Each mouse was assessed individually, but with the co-housed animals present in the home cage. Cage means were treated as the experimental unit, unless stated otherwise.

### Body temperature

Surface body temperature was determined non-invasively using an infrared thermometer (GT-101, Geratherm Medical AG, Geratal, Germany) by holding the thermometer over the back region of the mice (2–3 mm away). Temperature measurements were performed immediately at the end of each training session and again 15 minutes after completing the training session and transferring the mice back into their home cages. Each mouse was assessed individually, but with the co-housed animals present in the home cage. Cage means were treated as the experimental unit.

### Subcutaneous tracking chips and activity tracking

For movement tracking, a Radio Frequency Identification (RFID) Chip (ID162-PM) was implanted subcutaneously in the caudal abdominal region left of the Linea alba, using the IntelliCage® Injector (Cat.No. D-290000-IC-50705). The procedure was performed at the age of 5 weeks under short-term isoflurane anesthesia. The 2 x 2 mm wound channel was closed using a single staple with a 5/0 Monocryl suture. Cages were placed on Trafficage devices (TSE Systems, Berlin, Germany) to monitor activity of the animals inside the cages. Every cage was divided into 6 areas and each time an animal moved from one area to another was recorded as an event (S2 Fig). Measurements were analyzed using Microsoft Access. Co-housed mice were in the same experimental groups and tracked at the same time; cage means were treated as the experimental unit.

### Blood draws from the tail vein

Healthy untrained female C57Bl/6N mice not undergoing HCC induction (Table 1) were used to assess stress levels after blood draw from the tail vein. Mice were randomized to undergo blood draws either in the injection tunnel or a conventional restrainer (clear plastic tube without bedding, length 11 cm, inner diameter 3 cm, manufactured in-house; S6A Fig). In both cases the lateral tail vein was scored with a needle and one drop of blood collected into a microtube. Stress parameters were assessed immediately after releasing mice from the restrainer or the tunnel: pictures to assess MGS were taken on the Vetbed as soon as the mice exited the restrainer or tunnel; latency to interact was measured directly

**Table 1. Overview of experimental groups to evaluate effect of restraining method on stress levels after blood draws.**

| # | Disease model | Procedure | Training regimen | Restraining Method |
|---|---|---|---|---|
| 1 | Untreated (n=8) | Blood draw from tail vein | None | Conventional Restrainer |
| 2 | Untreated (n=7) | Blood draw from tail vein | None | Tunnel |

after placing the animals back into their home cages individually. Therefore, Individual mice were treated as the experimental units for stress parameters taken after blood draws. Two animals had to be excluded from the study, because blood draw was not successful on the first try and animals had to undergo restraining procedure twice, which could potentially falsify the results due to higher stress levels.

## Mouse grimace scale

The MGS was first introduced in 2010 to detect facial pain expression and categorization in mice [31]. Since then, it is a widely used parameter in pain and stress assessment [32]. For classification of animal welfare the MGS can be used as a scoring system for the characterization of reaction to pain by analyzing expression of facial muscles [32,33] and is a useful tool to determine the stress level in mice, e.g., after repeated anesthesia [34]. The scores were determined using photographs (n=2–3 per mouse) taken directly after the procedures before placing the animals back into their home cages. Therefore, individual animals were treated as the experimental unit in this case. Two observers with an experienced background on mice behavior scored three different facial expression parameters: Orbital tightening, nose bulge, and ear position. We did not assess whisker position and cheek bulge as this was challenging due to the dark fur color of our mice (a limitation already noticed in other animal welfare studies [35,36]). The parameters were scored on a three-point scale from 0 to 2 (0=not present, 1=moderately present, 2=obviously present). The three facial scores were then added up and averaged (mean MGS scores). Each animal was scored by both observers independently and the 2 MGS scores averaged to reach one final MGS score per animal.

## Pathological examination

We randomly selected 3 conventionally injected mice and 4 mice injected in the tunnel to be sampled for histopathological examination of the injection site. Mice had received the last intraperitoneal injection 72h before sampling. At sacrifice via isoflurane overdose, tissue samples of about 2 x 1cm including peritoneum, abdominal wall and skin were collected from the caudal abdominal region where mice had received injections and stored in 4% paraformaldehyde solution until further processing. Samples were paraffin embedded, cut into 5 µm sections and stained with hematoxylin and eosin at the Institute of Veterinary Pathology, Freie Universität Berlin. Histopathological examination was performed by an EBVS® European Specialist in Veterinary Pathology.

## Fur collection and Corticosterone analysis

Fur samples were collected by carefully cutting the hair on the back of the mice with fine scissors as close as possible to the skin. Samples were wrapped in aluminum foil and stored at room temperature in the dark to avoid light exposure. Steroid extraction and quantification were performed by *Dresden Lab Service GmbH* (Dresden, Germany) as previously described [37,38]. Briefly, the fur samples were washed by shaking them in 2.5mL isopropanol for 3min at room temperature. After drying for 12 hours, 5±0,5mg of non-pulverized hair was transferred to a 2ml centrifuge tube. After adding 50 µl of internal standard and 1,8mL Methanol, the sample was incubated for 18h. Thereafter, the sample was centrifuged at 10000rpm for 2min and 1,6ml of the supernatant was transferred to a new 2ml tube. Alcohol was evaporated at 50°C and under a constant stream of nitrogen. The dried samples were resuspended in water and analyzed via liquid chromatography–tandem mass spectrometry (LC-MS/MS) assay with atmospheric pressure chemical ionization (APCI) positive

mode on a Shimadzu LC-20AD HPLC unit with a Shimadzu SIL-20 AC autosampler and a Shimadzu CTO-20 AC column temperature oven.

## Statistical analysis

Statistical analysis was performed using GraphPad Prism 10. Due to the small sample sizes non-parametric tests were performed as a default. Comparison of two groups was performed using Wilcoxon matched-pairs signed rank test for paired data and Mann Whitney test for unpaired data. Comparison of more than two groups was performed using repeated measures Friedman test with Dunn's multiple comparisons test (comparing every mean to a control mean) or repeated measures 2-way ANOVA of ranked data followed by Šidák´s multiple comparison test, as indicated. P-values under 0.05 were considered significant. Effect size r was calculated by dividing the standardized test statistic (z) by the square root of the total sample size (N). Following Cohen´s rule of thumb we considered effect sizes <0.3 as small, between 0.3–0.5 as medium and >0.5 as large. Two mice had to be excluded from experiments due to technical issues during blood draw (see above), but no additional data points or outliers were removed during analysis.

## Results

### Latency to interact

As a first and very immediate parameter to determine training effect, we measured the latency to interact after every training session. As expected, the latency to interact was quite high and very variable in untrained mice not used to the experimenter (S3 Fig and Fig 4A and 4B training day 1). In mice without training, the latency to interact remained high even when the animals were tested repeatedly (S3A Fig), whereas a decrease was apparent in all training groups already after the first day of training (Fig 4A and 4B). This effect was sustained even after the beginning of frequent ip injections (between training day 5 and 6 for CCl$_4$ mice and between training day 17 and 18 for WD mice). While not all timepoints reached significance when compared to the first day of training, effect sizes were generally quite large (>1.0 in many cases; S3B Fig). Furthermore, in the CCl4 model we did not observe any differences between mice injected using the conventional scruffing method or the application tunnel (Fig 4A).

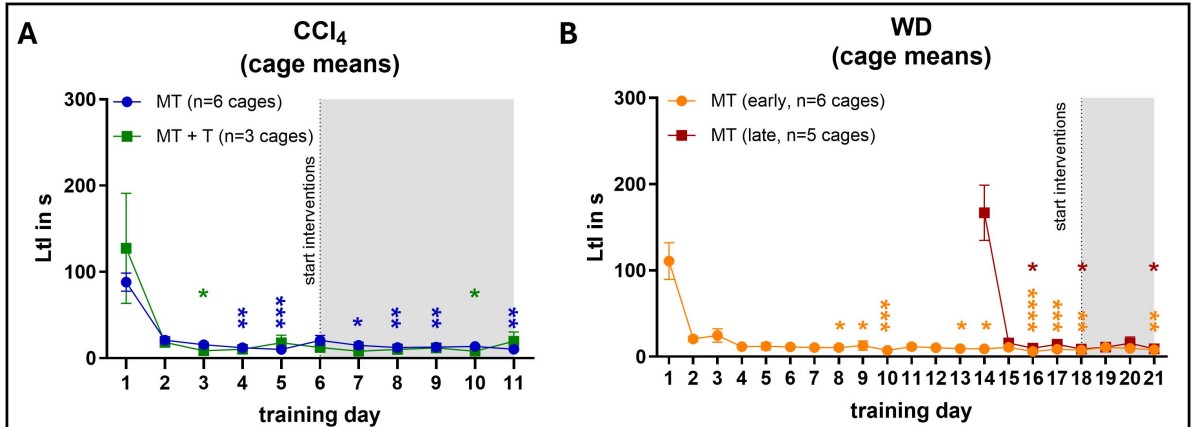

**Fig 4. Latency to interact.** Mice were subjected to the medical training regimen described in Fig 1 starting at the age of 6 weeks. After each training session the Latency to interact (Ltl) was recorded for every animal individually, but with co-housed animals present in the home cage. Cage means were treated as the experimental unit with the average Ltl of all mice in one cage as the datapoint. **(A)** Latency to interact of CCl$_4$ mice undergoing medical training only (MT) or medical training with tunnel injection (MT+T), **(B)** Latency to interact of WD mice receiving early (MT early) vs. late (MT late) medical training. Data are displayed as mean+/- SEM. Repeated Measures Friedman test with Dunn'smultiple comparisons test (all time-points compared to first day of training). *=p<0.05, **=p<0.01, ***=p<0.001, ****=p<0.0001.

In the WD model, we also compared mice that received an early basic training starting at the age of 6 weeks followed by a continuous training (WD + MT early) with mice that received a late training starting at the age of 22 weeks (WD + MT late) (Fig 4B). As described above, we observed a strong decrease in the latency to interact after the first training day in both groups, which did not increase again after the beginning of the ip injections, indicating that the age of the mice did not have an influence on the training effect.

## Body temperature

Next, we assessed the surface body temperature immediately and 15 min after each training session. We expected to observe a higher temperature in stressed mice due to the stress-induced hyperthermia response [39,40]. Hence, body temperature should be higher immediately after the training sessions as compared to 15 min after returning to their home cage and successful stress reduction via medical training should result in a lower surface body temperature over time. However, we observed the opposite in the CCl4 + MT group (Fig 5A). The temperature right after the training session was lower (mean minimum 35,33°C, maximum 35,61°C) compared to the temperature 15 minutes after (mean minimum 35,66°C, maximum 35,82°C). Of note, this difference was could be observed at every timepoint during the course of the experiment, but did not reach significance. The average body temperature – both immediately and 15 minutes after each session – did not change with increasing training sessions (i.e., with increasing age of the mice). Similar results were obtained for MT + T mice with minimal surface measurements of 34,8°C and maximum measurements of 35,69°C right after the training and a minimum of 35,35°C and a maximum of 35,86°C after 15 minutes recovery (Fig 5B). It is important to note that $CCl_4$ -treated mice were handled inside a biosafety cabinet with a ventilation system with high air flow due to the hazardous properties of $CCl_4$, which results in a rather cold surrounding atmosphere.

In comparison, mice of the WD-model (Fig 5C and 5D) showed no differences of surface body temperature right after (Mean WD early = 35,61 °C and WD late = 35,43°C) and 15 minutes after the medical training session (Mean WD early = 35,7 °C and WD late = 35,44°C). Those mice underwent medical training under the same conditions as their housing conditions (20–24°C). The animals had an increased body temperature over their training sessions and lifetime, and the WD early group displayed a temporary rise in temperature right after immunotherapy onset.

## Activity assessment after ip injections

In previous studies, we noticed that mice often show an apathic behavior directly after ip injections, especially after the injection of $CCl_4$. $CCl_4$ intoxication is known to cause symptoms like headaches, nausea and abdominal pain in humans [41], which may well explain the side-effects we observe in our mice: about 5–10 min after the injection, the animals start showing apathic behavior including closed eyes, curved backs, laid-back ears and above all, severely restricted movement for 30–60 minutes. Since we observed that this behavior seemed less pronounced when the mice were injected inside the novel injection tunnel, we assessed the activity of the mice after $CCl_4$ ip injections using subcutaneous RFID chips. For each mouse, the number of movements was collected over a period of three hours following ip injection either in the tunnel or using the conventional scruffing method for several (up to 19) time points per mouse (Fig 6). To generate a baseline, the same mice were tracked after cage cleaning and medical training sessions (S4A Fig). Since mice in the same experimental groups were housed together and assessed at the same time, we compared the cage means at each time point rather than the movements of individual mice. In general, all mice were less active after ip injections than after training sessions, but mice injected inside the application tunnel were more active after the injections than conventionally injected mice (Fig 6A). In addition, tunnel injected mice seemed to recover faster as well (Fig 6B). During the first hour after injection, the count of movements did not differ between both groups, however, during the second and third hour the tunnel injected group showed significantly more activity than conventionally injected mice. Both groups had a decreased movement again into the third hour after injection, possibly due to the handler having left them alone for quite some time by then, but tunnel injected mice were still more active than the conventionally injected group. Importantly, the positive

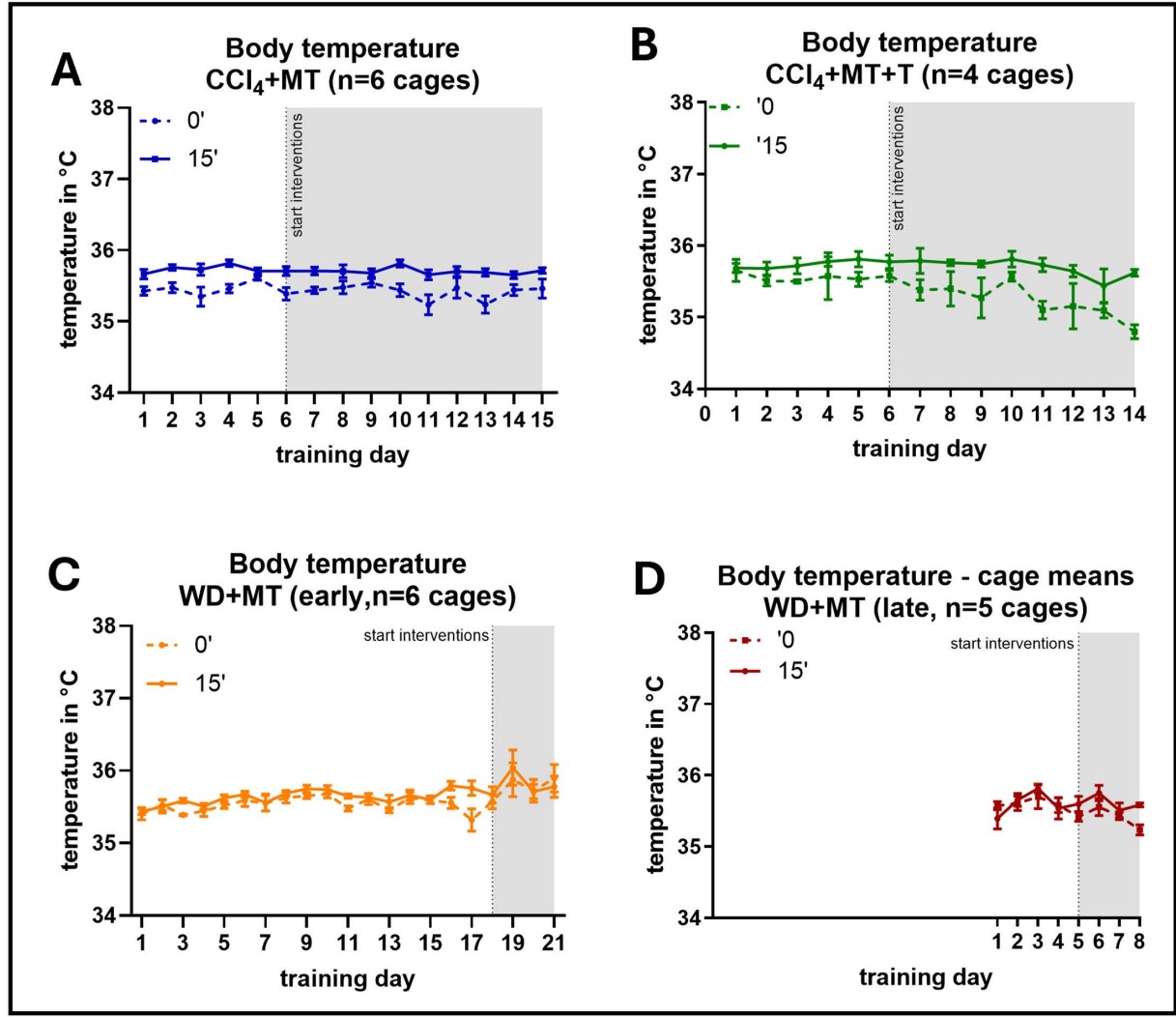

**Fig 5. Body temperature.** Body temperature measured immediately (0') and 15 minutest (15') after training sessions. Body temperature was measured for every animal individually, but with co-housed animals present in the home cage. Cage means () were treated as the experimental unit with the average body temperature of all mice in one cage as the datapoint. **(A)** CCl$_4$ mice with medical training (CCl4+MT), **(B)** CCl$_4$ mice with medical training and tunnel injection, **(C)** WD mice with early medical training (WD+MT early) and **(D)** WD mice with late medical training (WD+MT late). Data are displayed as mean +/- SEM. 2-way ANOVA of ranked data with Šidák´s post-test. *=p<0.05.

effect on the activity of the mice did not seem dependent on the injected substance, as we observed the same trend after ip injection of our immunotherapy drug (Flt3L, S4B and S4C Fig).

## Defecation and urination

We also recorded cases of urination and defecation during the injection procedure in two groups of mice, which received CCl$_4$ injections either conventionally or in the application tunnel. During the first 3 injection timepoints (i.e., the first 3 injections after completing the training regimen), we detected 10 cases of urination and 4 cases of defecation from conventionally injected mice (n=5) compared to no defecation and only one case of urination from mice injected inside the tunnel (Fig 6C).

 

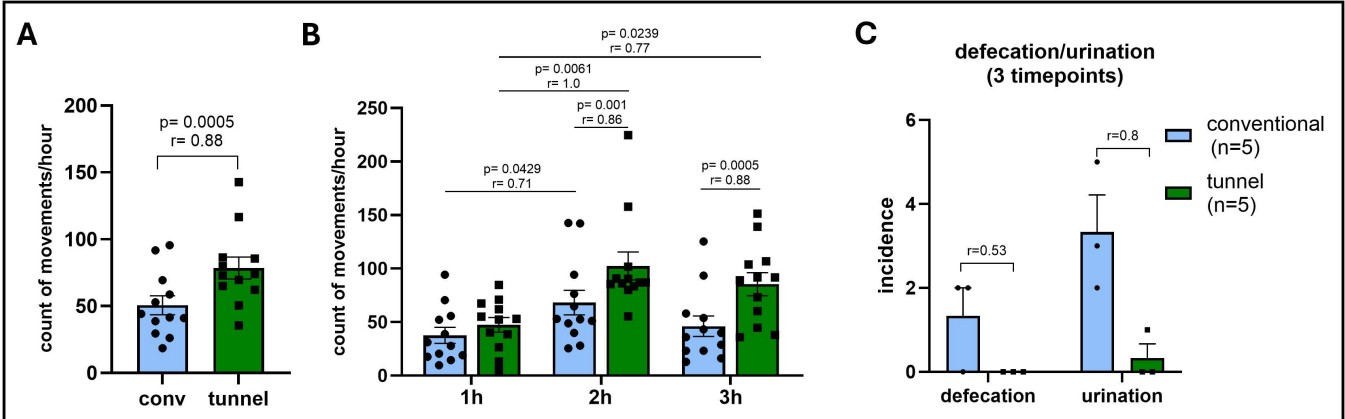

**Fig 6. Activity tracking after ip injections.** Mice were implanted a subcutaneous RFID chip in the abdominal region. After ip injections using either conventional scruffing or the novel injection tunnel, cages were placed on a traffic cage system and movements of the mice were recorded for a total of 3 hours after each injection. Because co-housed mice were tracked together, cage means were treated as the experimental until with the average count of movements of all mice in one cage per timepoint as the datapoint. Conventional: n = 10 mice in 2 cages; tunnel: n = 9 mice in 2 cages. **(A)** Count of movements after tunnel or conventional injection of CCl₄ measured over 3h at 12 different timepoints during the course of the experiment. Every data point represents the average count of movement per hour of all mice in the group at one experimental timepoint. Wilcoxon matched-pairs signed rank test; **(B)** Count of movements after tunnel or conventional injection stratified by the first three hours after CCl₄. Friedman test with Dunn's multiple comparisons test was used to compare individual timepoints within conventional or tunnel groups, Wilcoxon matched-pairs signed rank test was used to compare conventional vs. tunnel groups at each timepoint; **(C)** Incidence of urination and defecation during injection (each data point represents the sum of events in each group at 3 different experimental time points). Mann-Whitney test. Data are displayed as mean +/- SEM. r = effect size.

## Pathological examination

To ensure that the altered positioning of the animals during injection did not result in injuries to the application area we also examined the injection sites of 3 randomly selected conventionally and 4 randomly selected tunnel-injected mice histopathologically (S5 Fig). All mice showed moderate to severe inflammation of the skeletal muscles and, in some cases, seroma-like cystic cavities, which are likely the result of the regular and frequent injection needle puncture. However, in the pathological examination, no differences were observed between the two injection groups in a blinded assessment, suggesting that injection inside the tunnel did not cause any additional injuries.

## Blood draws

The encouraging results of the movement tracking and behavior of the mice during and after ip injections prompted us to also test the injection tunnel for blood draws from the tail vein using needle puncture. The existing restrainers to immobilize the mice during this procedure often cause signs of fear like vocalization, defecation and urination. Therefore, we assessed the stress level of mice after blood draw inside the restrainer in comparison to mice after blood draw using the application tunnel. For this purpose, we used healthy untrained mice in order to assess the effect of the tunnel without any confounders from the disease induction or the training. The latency to interact measured right after blood draw was significantly lower for the tunnel-handled mice (mean = 13,14 s) compared to restrainer-handled mice (mean = 41,5 s) (Fig 7A). We also noticed variation in facial expression and general habitus. Pictures taken right after the blood draw (S6 Fig) were assessed for orbital tightening, nose bulge and ear position on a categorical scale from 0 to 2. Tunnel-handled mice displayed a significantly lower MGS score compared to restrained mice (Fig 7B). Taken together, these results indicate that restraining mice in the tunnel also increases welfare during and after blood draws from the tail vein.

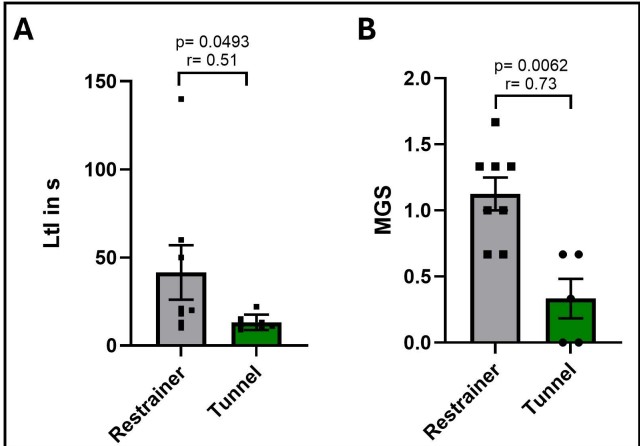

**Fig 7. Latency to interact and grimace scale after blood draw from the tail vein.** Mice were subjected to blood draws either inside a conventional restrainer (n = 8) or the novel application tunnel (n = 7). **(A)** Latency to interact (LtI) measured immediately after blood draw. Every animal was assessed individually with no other animals present in the home cage. Mann Whitney test. **(B)** Facial expression assessed by mouse grimace scale (MGS) immediately after blood draw. Mice were assessed individually before being returned to the home cage. Mann-Whitney test, Data are displayed as mean +/- SEM. r = effect size.

## Corticosterone accumulation in fur

As a long-term stress parameter, we assessed corticosterone accumulation in the fur at the age of 22 weeks (CCl$_4$ model) and 26 weeks (WD model) and compared corticosterone levels to 22-weeks old healthy untrained mice without any disease induction or interventions. We were not able to detect any significant differences in either model (S7A and S7B Fig), with a tendency towards lower corticosterone levels in WD mice in the early training group, which also received a continuous training throughout the whole experiment (for about 20 weeks). This may indicate that the medical training has a positive influence on the welfare of the mice even during regular housing without interventions. This effect was not observed in the late training group.

## Discussion

Our study shows that habituation of mice to interventions with medical training in combination with the use of a novel application tunnel reduces stress levels during frequent ip injections and blood draws. These effects were sustained throughout the course of the experiment and applied even after injection of substances that are known to have a negative effect on well-being themselves. Similar to what other studies have shown [29,42], a training frequency of 5 training days was sufficient for us to get the animals used to the handling procedure, e.g., for ip injections. Neither the timepoint of the training start within the experiment nor the beginning of the injections had an influence on the training success. As expected, measuring the latency to interact with the experimenter as an immediate stress parameter was very helpful to determine the effect of the training. Since this parameter decreased rapidly after only one training session and remained constant after that, it is possible that this change in approach behavior was supported by familiarity with the test from day 2 onwards, which would suggest that less training still maintains the same benefits. However, the same reduction could not be observed in mice that did not receive any training and were only assessed for the latency to interact on the same days as the training groups. Still, as each training session creates some level of stress on the animals, achieving the desired effect with as few training sessions as possible is highly desirable and would also save considerable lab resources (experimenter time). Nevertheless, we noticed that it is essential for the animals to be introduced to the application tunnel during the training program in order to facilitate them entering the tunnel without force. Therefore, the sessions for training

of the injection procedure should not be omitted. Otherwise, the time saved on training would be offset by the additional time needed to convince animals to enter the tunnel and the added stress of using force might negate the purpose of the training. In contrast to the latency to interact, the measurement of body temperature showed no changes over the course of the medical training sessions and rather seemed dependent on external influences as working under ventilation systems (resulting in lower surrounding temperature) seemed to distort the values strongly. The risk of temperature loss for mice with a relatively large surface area compared to their body weight is known [43]. Therefore, the reduced surface temperature observed in $CCl_4$ -treated animals was likely due to the surrounding atmosphere in the biosafety cabinet rather than stress-related. In addition, the increase in body temperature observed in WD animals over time was likely due to the diet-induced obesity and decreasing body surface to weight ratio [44] and the temperature peak in the WD early group right after immunotherapy seemed to be more influenced by immunotherapy itself than training effort [45]. Therefore, it might have been more informative to test both $CCl_4$ and WD mice in similar environmental conditions, however, this was not possible due to animal facility regulations (due to its hazardous properties, $CCl_4$ -treated mice had to be kept in a separate room and cages could only be opened under a biosafety cabinet to protect experimenters). Nevertheless, since we were still able to detect rather small temperature deviations (between 0 and 15 min after training in cold environments), surface measurements can generally be a good alternative to intrarectal measurements, which has been suggested by other studies as well [18,46]. It is important to keep in mind that there is a big variation in surface temperature depending on the location of measurement [47], which can falsify the results due to vasoconstriction in the periphery as a reaction to stress and pain. These issues could be circumvented by using infrared thermography, which is able to non-invasively capture the temperature of different body areas at the same time, or implantable transponder-based systems to measure core temperature [16,47]. However, these methods require more resources in terms of time, expertise and money and might not be available in every research environment. In this case, measuring the temperature in proximal areas and with a probe held close to the surface, as we performed here, leads to more accurate and consistent results compared to a contactless measurement [48].

In mice, handling procedures during laboratory routine itself, which includes lifting the mice up, cleaning and moving the cages, can already trigger fear and pain [5]. Invasive interventions, like injections, which are associated with harsh restraining are increasing those effects. It has been shown that ip injections with saline only can increase stress parameters like fecal corticosterone metabolites [47]. Catching mice by the tail can cause pain and fear and many studies show improved welfare when replacing tail handling by tunnel or cup handling [15,29] and this beneficial effect remain even when animals are subject to frequent interventions like ip injections [49,50]. In line with this, we observed that not only the method of catching and transferring mice, but also the type of restraining during injection appears to have a major impact. The measurement of activity patterns after ip injections showed that mice that experience less restraining injection had a significantly faster recovery period. Importantly, this effect seemed to be independent of the injected substance. Furthermore, we noticed that use of the application tunnel can also reduce stress in researchers, especially those inexperienced in handling and fixation of animals. Conventional restraining methods can lead to stress caused by fear of getting bitten while scruffing the animal, which is aggravated by failed attempts and repeated tries to achieve correct positioning of the mouse. Increased stress levels of the researchers transfer to the animals and heightens their fear, which makes restraining even more difficult. The animals included in this study were all handled by experienced handlers to exclude level of experience as a confounder. However, it is worth mentioning that many other researchers at our institution using the tunnel for intraperitoneal injection in their own studies, including some that were newly trained, reported decreased hesitation in approaching the animals and performing injections.

We also observed reduced signs of pain after blood draws from the tail vein inside the new application tunnel, which we assessed by taking the MGS immediately after the procedure. Mice handled in a conventional restrainer showed higher MGS compared to those, which experienced the same procedure less restrained in our application tunnel. This is in accordance with a previous study that demonstrated pain differences based on facial expression during ear-tagging depending

on the handling method (tunnel or tail). Tunnel handled mice showed significantly lower MGS after ear-tagging [51], suggesting that handling method plays an important role in the response of mice to stressful and painful procedures. Still, the experience of the scorers can lead to significant variability in MGS results [32] and needs to be taken into account. Already existing approaches in (half-)automated evaluation of MGS are promising tools for standardized assessment of facial expressions [52,53] and could allow the use by non- or less-experienced experimenters.

In our study, we could not measure any significant changes in the end-point stress parameter corticosterone in the fur of our mice. Neither medical training nor tunnel injection were able to lower corticosterone levels of the mice in the $CCl_4$ groups and only a tendency could be observed in the WD groups. It is unclear, whether this means that chronic stress levels in our mice were unchanged or whether corticosterone levels were influenced by any other experiment-related conditions. It is known that corticosterone levels show higher variability in grouped mice, potentially caused by hierarchy with the dominant animal in the cage showing the highest levels [23]. The frequent handling of the $CCl_4$ mice can lead to increased aggressiveness and dominance behavior, as the animals constantly have to fight out and defend their hierarchy when reassembling/relocating in their cages, which may cause higher variability of corticosterone levels in individual mice. Additionally, it should be noted that sampling the fur may be prone to distortion of data through circumstances such as mice grooming each other in groups and transferring the corticosterone-containing saliva of one mouse to another. This problem could be circumvented by measuring corticosterone metabolites in urine or feces, but collection of these samples can lead to increased acute stress levels due to single-housing or use of special metabolic cages without nesting material or housing and therefore also skew results [54]. In addition, corticosterone is underlying a circadian rhythm with a peak in the evening hours during the awakening response of the nocturnal animals [19], which further complicates standardized collection as fecal droppings in the home cage would need to be from the same time of day. This would require cage changes directly before the collection period, which could also increase short-term stress levels in the animals. Our results are in line with a recent study in rats that challenged the general concept of hair corticosterone as a representative of chronic stress and concluded that glucocorticoid levels in fur can only indicate recent stress as they diffuse out of hairs again within a few days after a stress stimulus [55]. This might explain why our end-point measurements were not able to show any differences between the groups. Further studies with more frequent sampling during the course of experiments would be needed to confirm this.

In conclusion, our study shows that the medical training program is able to habituate the mice to human contact and reduce handling-associated stress during interventions. In addition, the use of the new application tunnel seems to significantly mitigate the negative influence of procedures like injections and blood draws and further reduced signs of stress, fear and pain. Therefore, our medical training protocol provides a good concept to increase and monitor animal welfare in the context of long-term experiments with frequent ip injections, which can easily be adapted depending on the experimental design. The application tunnel is easily customizable to individual needs as well. It can be printed on basic 3D-printer models using low-cost UV-cured resin making it accessible to a wide scientific community even in limited resource settings. Given the large number of rodents used in biomedical research worldwide, our application tunnel has the potential to make a significant and lasting impact on laboratory animal welfare.

## Supporting information

**S1 Table. Overview of experimental groups.**
(PDF)

**S2 Table. Overview of the basic training program.**
(PDF)

**S1 Fig. Technical diagram of the injection tunnel.**
(JPG)

**S2 Fig. Traffic cage system.**
(JPG)

**S3 Fig. Latency to interact in mice not subjected to medical training.**
(JPG)

**S4 Fig. Activity tracking after ip injections or cage cleaning.**
(JPG)

**S5 Fig. Representative H&E pictures of abdominal wall after injection.**
(JPG)

**S6 Fig. Pictures used for mouse grimace scale assessment after blood draw.**
(JPG)

**S7 Fig. Corticosterone accumulation in fur.**
(JPG)

**S1 Checklist. ARRIVE checklist.**
(PDF)

## Acknowledgments

We thank our animal welfare officer Dr. Juliane Unger for her support in designing and implementing this study and all members of the Charité 3R Refinement Taskforce for helpful discussion.

## Author contributions

**Conceptualization:** Bernhard Hiebl, Linda Hammerich.

**Data curation:** Anne Schlutt, Katarina Riesner, Merle Kochan, Ann-Kathrin Meß, Denise Jahn, Isabella Lurje, Wiebke Werner, Olivia Kershaw, Linda Hammerich.

**Formal analysis:** Anne Schlutt, Katarina Riesner, Olivia Kershaw.

**Funding acquisition:** Linda Hammerich.

**Investigation:** Anne Schlutt, Isabella Lurje, Wiebke Werner, Linda Hammerich.

**Methodology:** Felix Heymann.

**Project administration:** Linda Hammerich.

**Supervision:** Frank Tacke, Bernhard Hiebl, Linda Hammerich.

**Visualization:** Anne Schlutt, Linda Hammerich.

**Writing – original draft:** Anne Schlutt, Linda Hammerich.

**Writing – review & editing:** Katarina Riesner, Merle Kochan, Ann-Kathrin Meß, Denise Jahn, Isabella Lurje, Wiebke Werner, Olivia Kershaw, Frank Tacke, Bernhard Hiebl, Linda Hammerich.

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
