## [Decision Letter · Decision Letter 0]

12 Feb 2026

PONE-D-26-00796A novel application tunnel in combination with medical training reduces stress induced by frequent intraperitoneal injections and blood sampling in micePLOS One

Dear Dr. Hammerich,

Thank you for submitting your manuscript to PLOS ONE. After careful consideration, we feel that it has merit but does not fully meet PLOS ONE’s publication criteria as it currently stands. Therefore, we invite you to submit a revised version of the manuscript that addresses the points raised during the review process.

Please see the detailed comments given by the two independent reviewers below.

We look forward to receiving your revised manuscript.

Kind regards,

I Anna S Olsson, Ph.D.

Academic Editor

PLOS One

Journal Requirements:

2. To comply with PLOS One submissions requirements, in your Methods section, please provide additional information regarding the experiments involving animals and ensure you have included details on (1) methods of sacrifice, (2) methods of anesthesia and/or analgesia, and (3) efforts to alleviate suffering.

4. In the online submission form you indicate that your data is not available for proprietary reasons and have provided a contact point for accessing this data. Please note that your current contact point is a co-author on this manuscript. According to our Data Policy, the contact point must not be an author on the manuscript and must be an institutional contact, ideally not an individual. Please revise your data statement to a non-author institutional point of contact, such as a data access or ethics committee, and send this to us via return email. Please also include contact information for the third party organization, and please include the full citation of where the data can be found.

5. We note you have included a table to which you do not refer in the text of your manuscript. Please ensure that you refer to Table 3 in your text; if accepted, production will need this reference to link the reader to the Table.

Reviewers' comments:

Reviewer's Responses to Questions

**Comments to the Author**

1. Is the manuscript technically sound, and do the data support the conclusions?

Reviewer #1: No

Reviewer #2: Partly

2. Has the statistical analysis been performed appropriately and rigorously? 

Reviewer #1: No

Reviewer #2: I Don't Know

3. Have the authors made all data underlying the findings in their manuscript fully available?

Reviewer #1: No

Reviewer #2: Yes

4. Is the manuscript presented in an intelligible fashion and written in standard English?

Reviewer #1: Yes

Reviewer #2: Yes

5. Review Comments to the Author

Reviewer #1: The authors report on a novel tunnel designed for routine applications (injections and blood draws) that can reduce handling stress.

I have some concerns regarding statistical analyses.

1. The authors correctly use cage as the experimental unit by averaging measurements within cages (avoiding pseudoreplication), however, the sample sizes appear quite small (N=3-8 cages per group) with unequal group sizes.

The authors state they used the Shapiro-Wilk test to determine whether to use parametric or non-parametric tests. Shapiro-Wilk test has low power to check for normality when sample size is small. With small sample sizes across the study (<10), non-parametric alternatives should be used by default.

2. All statistical results should be clearly reported in the Results section. Given the small sample sizes, it would be better to focus on effect sizes and confidence intervals for interpreting the biological significance of results.

3. According to Table 1, the number of cages per treatment group ranged from 2 to 6, and the number of mice per cage was also highly variable (from 1 to 6). So, for example, looking at Table 1, measures for corticosterone were taken from a singly housed mouse in C1, and averaged from 5 mice in cage C6. When cage means are calculated from different numbers of mice, each cage mean will probably have different variance. I am not sure treating a single mouse is equivalent to the mean of 5 or 6 mice. ANOVA and t-tests assume equal variance across groups, which is probably violated here.

4. Line 298-301 – the authors report that because animals were tested individually after injections, individual mice were treated as individual units for statistical analysis. Mice from the same home-cage should not be treated as independent even if they were tested individually. Experimental unit should be cage. Mixed models can be used for analysis, where mouse nested in cage is used as a random factor.

5. I have an additional question about voluntary interaction measurements. The authors measure latency to approach the experimenter’s hand. The literature that they cite (Henderson et al 2020, Gouveia et al 2013), that also conducted voluntary interaction tests however, measure differences in interaction time (% time spent interacting with the experimenter’s hand during 60s). Can the authors explain why they used a different metric here (latency to start interacting) as the papers they cited?

Also, in the first part of the experiment you measure latency to interact in group housed mice, but in the second experiment you measure it on each individual mouse before it was placed back in the home cage. I don’t think either method is better, but the difference in measuring interaction should be more explicitly stated in the discussion as it could have affected the results.

Some additional minor points:

Abstract: the strain and sex of animals tested should be mentioned in the abstract.

Line 57: It should be clearly stated that the latency to interact with experimenter decreased for both treatment groups, or there was no treatment effect.

Line 58: Change “normal behaviour” to “normal activity levels” as you did not measure behaviour but activity (as mentioned in Results, line: 434).

Line 55: The authors mention that body surface temperature was measured, but the result is not mentioned in the abstract. All outcome measures that were assessed should be mentioned in the abstract.

Lines 119-128: this paragraph should go at the beginning of the discussion.

Line 141: It is not clear from the times stated (4pm and 7pm) if experiments were conducted during the light or dark phase.

Reviewer #2: Schlutt and colleagues propose a refinement strategy for intraperitoneal injections and blood sampling based on the use of a tunnel in combination with a training protocol. The manuscript addresses an important issue in the context of the 3Rs by proposing a refined strategy for routine procedures in animal facilities. As such, I consider it to be an important addition to the knowledge base on the 3Rs.

Nevertheless, there are several errors and points that require clarification. Specific comments on the different sections are provided below.

Introduction

• On lines 87–88, the authors refer to a “higher risk of painful mistakes due to wrong restraining or application.” Please clarify what is meant by these “painful mistakes.”

• Throughout the manuscript, the term “medical training” is used to describe the training protocol/program. I suggest using a different term, as this wording may be misleading.

• On line 117 and in most of the manuscript, the term “blood draws” is used, whereas the title uses “blood sampling.” I recommend standardizing the terminology throughout the manuscript.

• The Introduction would benefit from a more thorough review of previously developed refinement methodologies for substance administration in laboratory mice.

Materials & Methods

There is a recurrent lack of detailed information regarding the animals and their housing conditions. In accordance with the ARRIVE guidelines and reproducibility standards, I strongly recommend including the following information:

• Age range of the animals used in the study;

• Details of the cage type (e.g., dimensions, manufacturer);

• Details of the diet (e.g., manufacturer, treatment, product code);

• Details of water provision (e.g., water bottles vs. automated watering system, treatment);

• Details of nesting, bedding, and environmental enrichment (amount, manufacturer, product code).

• Line 153 – Please provide more details on the Western-style diet (e.g., percentage of fat, sucrose, protein, moisture content).

• Lines 159–160 – The authors state that animal “welfare and health status” were assessed daily. Please describe this procedure. Was this a daily visual health check?

• Lines 161–165 – The authors state that “death was not a likely outcome or planned experimental endpoint.” Please clarify whether “death” refers to spontaneous death or euthanasia. Death should be prevented through the application of humane endpoints and should not be considered an endpoint in itself. Moreover, all animals undergoing procedures should be monitored using predefined criteria and humane endpoints at all stages of the experiment.

• Line 166 – Humane endpoints included weight loss >20%. Why was body condition scoring not included? In oncological models with internal tumors, body condition score is generally more sensitive than body weight for assessing overall condition (see OBSERVE guidelines). Given that this is a refinement-focused manuscript, this omission should be justified.

• Line 167 – Tumor diameter is listed as a humane endpoint. How was tumor size measured?

• Table 1 – Table 2 presents an overview of experimental groups, assessed parameters, and the number of animals per cage. This table raises several questions:

o Why were not all parameters assessed in all experimental groups? For example, tracking was assessed only in the CCl₄ groups with medical training and tunnel handling, as well as in the conventional group.

o Why did the number of animals per cage vary from 1 to 6? Could this variation have influenced parameters such as corticosterone levels?

• Lines 228–232 – The description of the “conventional method” for intraperitoneal injections does not correspond to the method typically described for scruffing, nor does it match the description in the cited reference. In particular, the method of immobilizing the animal prior to scruffing is unclear. Conventionally, immobilization is achieved by securing the tail and allowing the mouse to grasp the cage grid with its forepaws. Please justify the choice of the described method. Additionally, given that this is a refinement-focused study, the authors should acknowledge that a more refined and widely accepted scruffing method—the three-finger method, which avoids pressure on the neck area—exists and has been described (see Labitt et al., 2021), as illustrated in Figure 2E.

• Line 266 – A space is missing between “interact” and “as.”

• Lines 270–271 – It is stated that “each mouse was assessed individually, but with the co-housed animals present in the home cage.” Why was this assessment not performed in complete isolation? Could dominance hierarchies or social interactions have influenced the results?

• Body temperature measurements (lines 275–280) – Why was the dorsal region selected for temperature assessment? As infrared thermometers measure surface temperature, contact with the handler’s hands during scruffing or proximity to the tunnel could have influenced the readings.

• Subcutaneous tracking chips and movement tracking:

o Why were RFID chips implanted in the caudal abdominal region, left of the linea alba? This placement is not recommended (see Kahnau, 2026) and is close to the intraperitoneal injection site, potentially increasing discomfort compared with dorsal subcutaneous implantation.

o At what age were the transponders implanted?

o Was analgesia provided?

• Blood sampling from the tail vein:

o Please provide more details on the “conventional restrainer.”

o Does the description provided encompass the entire procedure? Were any measures taken to promote vasodilation?

• Pathological examination – Please describe the euthanasia method used. How long after the final intraperitoneal injection was euthanasia performed?

Results

• Line 372 – A space should be inserted between “interact” and “after,” and between “interact” and “was.”

• Lines 372–373 – The authors state that “latency to interact was quite high and very variable in untrained mice,” but these data are not shown in the figure.

• Line 381 – A space should be inserted between “interact” and “after.”

• Figure 4 – This figure would be more informative if it also included data from untrained mice.

• Line 426 – I suggest replacing “Behavioral assessment” with “Activity assessment,” as no specific behaviors were scored.

• Lines 473–480 (Pathological examination) – Were any additional procedures performed to confirm the absence of tissue damage resulting from the modified technique?

• Lines 488–489 – The authors state that “healthy untrained mice” were used to avoid confounding effects of disease induction. Why was this approach not applied to the other experimental groups?

• Was the potential effect of the experimenter on the results evaluated or controlled?

Discussion

• Lines 549–554 – The discussion of temperature assessment methods is unclear and should be rephrased. Although the authors refer to different methodologies, they used a contactless infrared thermometer (measurements taken at 2–3 mm from the skin). The discussion should also address other non-invasive and accurate temperature assessment methods, such as thermography, as well as invasive alternatives like implanted microchip transponders.

• Lines 571–573 – The authors state that only experienced handlers were included in the study but then mention that newly trained researchers reported decreased hesitation. This sentence should be rephrased for clarity.

• Line 581 – The authors discuss the influence of observer experience on Mouse Grimace Scale (MGS) scores. Were the observers in this study experienced? How many scorers were involved?

• Lines 597–600 – The authors state that a limitation of fecal corticosterone analysis is the need for metabolic cages. This is incorrect, as fecal samples can be collected directly from the home cage.

• Lines 601–602 – Circadian rhythm is mentioned as a limitation of fecal corticosterone analysis; however, this can be mitigated by collecting samples consistently at the same time of day.

• Lines 602–607 – Given these known limitations, please justify the choice to quantify corticosterone in fur.

6. PLOS authors have the option to publish the peer review history of their article (what does this mean?). If published, this will include your full peer review and any attached files.

Reviewer #1: No

Reviewer #2: No

---

## [Author Response · Author response to Decision Letter 1]

17 Mar 2026

Thank you for evaluating our manuscript PONE-D-26-00796.

Please find our answers to the editor´s and reviewers´ comments below:

We have formatted the manuscript to adhere to all requirements.

2. To comply with PLOS One submissions requirements, in your Methods section, please provide additional information regarding the experiments involving animals and ensure you have included details on (1) methods of sacrifice, (2) methods of anesthesia and/or analgesia, and (3) efforts to alleviate suffering.

Sacrifice was not an endpoint for this study, except for pathological examination, we have added the method of sacrifice in the respective methods section. Anesthesia/Analgesia were not used in this study and methods to alleviate suffering are described (training program, health checks, improved injection procedure).

All data has been prepared to be uploaded to the OSF platform upon acceptance and will be available under the same link as the files for manufacturing the tunnel (view-only link in supplementary methods).

4. In the online submission form you indicate that your data is not available for proprietary reasons and have provided a contact point for accessing this data. Please note that your current contact point is a co-author on this manuscript. According to our Data Policy, the contact point must not be an author on the manuscript and must be an institutional contact, ideally not an individual. Please revise your data statement to a non-author institutional point of contact, such as a data access or ethics committee, and send this to us via return email. Please also include contact information for the third party organization, and please include the full citation of where the data can be found.

This was related to the process of protecting the tunnel via a utility model. This situation has changed in the meantime and all data can be made available now (see above). We removed this statement from the respective section.

5. We note you have included a table to which you do not refer in the text of your manuscript. Please ensure that you refer to Table 3 in your text; if accepted, production will need this reference to link the reader to the Table.

Table 3 is now referenced in the text.

This did not apply.

Reviewer #1: The authors report on a novel tunnel designed for routine applications (injections and blood draws) that can reduce handling stress.

I have some concerns regarding statistical analyses.

1. The authors correctly use cage as the experimental unit by averaging measurements within cages (avoiding pseudoreplication), however, the sample sizes appear quite small (N=3-8 cages per group) with unequal group sizes.

The authors state they used the Shapiro-Wilk test to determine whether to use parametric or non-parametric tests. Shapiro-Wilk test has low power to check for normality when sample size is small. With small sample sizes across the study (<10), non-parametric alternatives should be used by default.

Thank you for pointing this out. We have recalculated all statistics using non-parametric tests. Details can be found in the methods section. Please note: since Prism does not provide a to analyze non-parametric data corresponding to a 2-way ANOVA, we transformed the data into ranks and performed a 2-way ANOVA on those ranks based on Scheirer et al, 1976 (this only applied to the body temperature measurements).

2. All statistical results should be clearly reported in the Results section. Given the small sample sizes, it would be better to focus on effect sizes and confidence intervals for interpreting the biological significance of results.

Thank you for suggesting this. We have calculated the effect sizes for all comparisons (described in methods) and report them either directly in the figures together with the p-values or as a supplementary table (in the case of Figure 4) for clarity. The majority of effect sizes could be considered large (>0.5).

3. According to Table 1, the number of cages per treatment group ranged from 2 to 6, and the number of mice per cage was also highly variable (from 1 to 6). So, for example, looking at Table 1, measures for corticosterone were taken from a singly housed mouse in C1, and averaged from 5 mice in cage C6. When cage means are calculated from different numbers of mice, each cage mean will probably have different variance. I am not sure treating a single mouse is equivalent to the mean of 5 or 6 mice. ANOVA and t-tests assume equal variance across groups, which is probably violated here.

Please see previous answer regarding statistics above, all statistics have been recalculated with non-parametric tests.

4. Line 298-301 – the authors report that because animals were tested individually after injections, individual mice were treated as individual units for statistical analysis. Mice from the same home-cage should not be treated as independent even if they were tested individually. Experimental unit should be cage. Mixed models can be used for analysis, where mouse nested in cage is used as a random factor.

This statement was related to blood draws, not injections. In the injection groups, all mice were tested individually, but with the co-housed animals present in the home cage, and we always used cages as the experimental unit. However, in case of the blood draws, mice were assessed individually without the co-housed animals being and without coming into contact with the co-housed animals between procedure and measurement of the stress parameters. Therefore, individual mice were treated as the experimental unit, equivalent to the general approach with regards to other anxiety tests (eg. open field), where mice are group-housed, but assessed individually without other mice present.

5. I have an additional question about voluntary interaction measurements. The authors measure latency to approach the experimenter’s hand. The literature that they cite (Henderson et al 2020, Gouveia et al 2013), that also conducted voluntary interaction tests however, measure differences in interaction time (% time spent interacting with the experimenter’s hand during 60s). Can the authors explain why they used a different metric here (latency to start interacting) as the papers they cited?

Thank you for catching this mistake. We indeed cited the wrong papers for this method. Our approach followed the study of Hohlbaum et al (2020), who measured “latency to first voluntary interaction with the experimenter’s hand” to assess “anxiety-related behavior in anticipation of handling”. We have replaced the original citations with the correct one.

Also, in the first part of the experiment you measure latency to interact in group housed mice, but in the second experiment you measure it on each individual mouse before it was placed back in the home cage. I don’t think either method is better, but the difference in measuring interaction should be more explicitly stated in the discussion as it could have affected the results.

The latency to interact was also measured inside the home cage after blood sampling, however without the cohoused mice being present. This is also stated in the respective method section. We decided to use this strategy, because the pictures for MGS had to be recorded as immediately as possible after the blood draws, ie. on the Vetbed right after releasing the mice from the tunnel or restrainer. Releasing the mice back into the home cage would have made it challenging to take a picture suitable to assess the MGS. Therefore, we also wanted to record the latency to interact individually instead of in cage groups.

Some additional minor points:

Abstract: the strain and sex of animals tested should be mentioned in the abstract.

This has been added

Line 57: It should be clearly stated that the latency to interact with experimenter decreased for both treatment groups, or there was no treatment effect.

This has been added.

Line 58: Change “normal behaviour” to “normal activity levels” as you did not measure behaviour but activity (as mentioned in Results, line: 434).

Thank you for suggesting this. We have changed this throughout the manuscript

Line 55: The authors mention that body surface temperature was measured, but the result is not mentioned in the abstract. All outcome measures that were assessed should be mentioned in the abstract.

This has been added to the abstract

Lines 119-128: this paragraph should go at the beginning of the discussion.

We combined this paragraph with the first paragraph of the discussion.

Line 141: It is not clear from the times stated (4pm and 7pm) if experiments were conducted during the light or dark phase.

We have added this information

Reviewer #2: Schlutt and colleagues propose a refinement strategy for intraperitoneal injections and blood sampling based on the use of a tunnel in combination with a training protocol. The manuscript addresses an important issue in the context of the 3Rs by proposing a refined strategy for routine procedures in animal facilities. As such, I consider it to be an important addition to the knowledge base on the 3Rs.

Nevertheless, there are several errors and points that require clarification. Specific comments on the different sections are provided below.

Introduction

• On lines 87–88, the authors refer to a “higher risk of painful mistakes due to wrong restraining or application.” Please clarify what is meant by these “painful mistakes.”

We have rephrased this sentence for clarity

• Throughout the manuscript, the term “medical training” is used to describe the training protocol/program. I suggest using a different term, as this wording may be misleading.

In veterinary medicine the term medical training is used for training of animals to voluntarily cooperate with medical procedures and examinations with the goal to reduce or eliminate the need for physical restraint and thereby reduce stress. We find this definition very fitting for the purpose of our training and would therefore like to keep this wording.

• On line 117 and in most of the manuscript, the term “blood draws” is used, whereas the title uses “blood sampling.” I recommend standardizing the terminology throughout the manuscript.

This has been changed

• The Introduction would benefit from a more thorough review of previously developed refinement methodologies for substance administration in laboratory mice.

Thank you for this suggestion. We have now added this to the introduction.

Materials & Methods

There is a recurrent lack of detailed information regarding the animals and their housing conditions. In accordance with the ARRIVE guidelines and reproducibility standards, I strongly recommend including the following information:

• Age range of the animals used in the study;

• Details of the cage type (e.g., dimensions, manufacturer);

• Details of the diet (e.g., manufacturer, treatment, product code);

• Details of water provision (e.g., water bottles vs. automated watering system, treatment);

• Details of nesting, bedding, and environmental enrichment (amount, manufacturer, product code).

We have added the requested information to the methods section

• Line 153 – Please provide more details on the Western-style diet (e.g., percentage of fat, sucrose, protein, moisture content).

We have added this information in the methods

• Lines 159–160 – The authors state that animal “welfare and health status” were assessed daily. Please describe this procedure. Was this a daily visual health check?

Yes, visual health checks were performed daily. We have added this in the methods section

• Lines 161–165 – The authors state that “death was not a likely outcome or planned experimental endpoint.” Please clarify whether “death” refers to spontaneous death or euthanasia. Death should be prevented through the application of humane endpoints and should not be considered an endpoint in itself. Moreover, all animals undergoing procedures should be monitored using predefined criteria and humane endpoints at all stages of the experiment.

Thank you for pointing out that this was not clearly described. Neither spontaneous death nor euthanasia were a likely or planned outcome in this study. We agree, that death should be prevented through the application of humane endpoints, which was the case in our study. We have clarified these points in the respective section.

• Line 166 – Humane endpoints included weight loss >20%. Why was body condition scoring not included? In oncological models with internal tumors, body condition score is generally more sensitive than body weight for assessing overall condition (see OBSERVE guidelines). Given that this is a refinement-focused manuscript, this omission should be justified.

We thank the reviewer for pointing this out. In fact, body condition scoring was included in our humane endpoints as we followed the OBSERVE guidelines for our study, but we forgot to mention this in the manuscript. We included weight loss as a separate endpoint independent of the body condition score, because the latter is often difficult to assess in the western-diet fed mice. Due to the obesity, a weight loss of 20% does not necessarily lead to a body condition score of 2 or lower, but is considered a humane endpoint as per local animal welfare regulations. Therefore, both parameters were considered as separate humane endpoints. We have now clarified this in the manuscript.

• Line 167 – Tumor diameter is listed as a humane endpoint. How was tumor size measured?

Tumor diameters were determined via magnetic resonance imaging

• Table 1 – Table 2 presents an overview of experimental groups, assessed parameters, and the number of animals per cage. This table raises several questions:

o Why were not all parameters assessed in all experimental groups? For example, tracking was assessed only in the CCl₄ groups with medical training and tunnel handling, as well as in the conventional group.

The purpose of the movement tracking was to prove that our novel injection technique, which consists of both training and use of the tunnel, improves the welfare of the mice after injections. As we have also described in the discussion, we noticed that it is essential for the animals to be introduced to the injection tunnel during the training program in order to facilitate them entering the tunnel without force. Otherwise, trying to convince the animals to enter the tunnel of forcefully placing them inside the tunnel would again result in additional stress for the animals. Therefore, we did not see benefit of assessing either part of the method (training vs. use of the tunnel) by itself.

o Why did the number of animals per cage vary from 1 to 6? Could this variation have influenced parameters such as corticosterone levels?

Animal numbers differed between cages, because animals entered the experiments at the age of 2 weeks (ie. before

---

## [Decision Letter · Decision Letter 1]

20 Apr 2026

PONE-D-26-00796R1A novel application tunnel in combination with medical training reduces stress induced by frequent intraperitoneal injections and blood draws in mice

PLOS One

Dear Dr. Hammerich,

Thank you for submitting your manuscript to PLOS ONE. After careful consideration, we feel that it has merit but does not fully meet PLOS ONE’s publication criteria as it currently stands. Therefore, we invite you to submit a revised version of the manuscript that addresses the points raised during the review process.

I acknowledge that all reviewer comments have met. My own careful reading of the manuscript show a number of minor inconsistencies, as listed below, which need addressing before the paper can be accepted.

Introduction:

First sentence: I don’t think one can back up the claim that rodents are more indispensable than other animal species! They are more commonly used but you are stating that explicitly and with a reference later on in the same paragraph.

Line 121-124 The last sentence summarizes the results and should therefore not be part of the Introduction

Line 159 “All male mice born in our facility” – is it really true that all male animals born in the entire facility underwent this protocol? Do you have an entire mouse facility dedicated to this line of research?

Tables 1 and 2 take up several pages and they present information in a level of detail that is more suitable for the Supplementary materials section than for the article proper. Please move these tables to the Supplementary materials, and include only a brief summary of them in the Materials and Methods section.

Line 270 Latency to approaching a human hand is a relevant parameter to measure in this study, but I don’t think that the main issue here is the stress effect on social interaction. Social interaction is not an appropriate description for how mice react to human handling. Animals interact socially with members of their own species, and may also develop a social relationship to individuals of other species (e.g. dogs, cats, horses, farm animals, rats) but that level of relationship is rarely established with a lab mouse. The human hand is the source of stress and discomfort in handling, and the mice are likely to associate the hand with the negative experience of being handled. Mice who have experienced less stressful handling are likely to show a shorter latency to approach the hand (do the mice really interact with the hand?). Please consider this also elsewhere in the manuscript where this concept is addressed.

Line 333 Please avoid starting a sentence with a digit.

Lines 416-417 What does “No remarkable changes were detectable depending on training success or age” mean? Please reword.

Lines 433-434 “Cage means (average body temperature of all mice in one cage) were treated as the experimental unit.”. I think most readers will understand what you mean, but to use precise and  correct terminology, the cage is the experimental unit, whereas the average body temperature of all mice in a cage is the datapoint.

Lines 512-518 How can you measure social interaction when the mouse is alone in the cage with no other mice present?

Line 566 “variation”, not “variety”

Line 577 “increasing”, not “maximizing”

Lines 578-580 Please reword. I suggest “many studies show improved welfare when replacing tail handling by tunnel or cup handling [15, 29] and this beneficial effect remain even when animals are subject to frequent interventions like ip injections.

Lines 616-617 I have never seen the word falsify being used in this way. In science, the word falsify stands for either the formal testing of hypothesis, or for researcher manipulation of data. Please reword. Also, you refer to “by various circumstances” but then only give one example.

We look forward to receiving your revised manuscript.

Kind regards,

I Anna S Olsson, Ph.D.

Academic Editor

PLOS One

**Journal Requirements:**

Reviewers' comments:

Reviewer's Responses to Questions

**Comments to the Author**

1. If the authors have adequately addressed your comments raised in a previous round of review and you feel that this manuscript is now acceptable for publication, you may indicate that here to bypass the “Comments to the Author” section, enter your conflict of interest statement in the “Confidential to Editor” section, and submit your "Accept" recommendation.

Reviewer #1: All comments have been addressed

Reviewer #2: All comments have been addressed

2. Is the manuscript technically sound, and do the data support the conclusions?

Reviewer #1: Partly

Reviewer #2: Yes

3. Has the statistical analysis been performed appropriately and rigorously? 

Reviewer #1: Yes

Reviewer #2: I Don't Know

4. Have the authors made all data underlying the findings in their manuscript fully available?

Reviewer #1: No

Reviewer #2: Yes

5. Is the manuscript presented in an intelligible fashion and written in standard English?

Reviewer #1: Yes

Reviewer #2: Yes

6. Review Comments to the Author

Reviewer #1: (No Response)

Reviewer #2: (No Response)

7. PLOS authors have the option to publish the peer review history of their article (what does this mean?). If published, this will include your full peer review and any attached files.

Reviewer #1: No

Reviewer #2: No

---

## [Author Response · Author response to Decision Letter 2]

21 Apr 2026

Thank you for evaluating our manuscript PONE-D-26-00796R1.

Please find our answers to your comments below:

Introduction:

First sentence: I don’t think one can back up the claim that rodents are more indispensable than other animal species! They are more commonly used but you are stating that explicitly and with a reference later on in the same paragraph.

We removed “especially rodents” from the first sentence as we agree that this applies to all laboratory animal species.

Line 121-124 The last sentence summarizes the results and should therefore not be part of the Introduction

This sentence was removed.

Line 159 “All male mice born in our facility” – is it really true that all male animals born in the entire facility underwent this protocol? Do you have an entire mouse facility dedicated to this line of research?

We acknowledge that this sentence may be confusing. Our mice are kept in a larger facility shared by many research groups, where we keep our own breedings since we need the pups at 14 days of age (ie. before weaning). All male mice born in the breeding cages of our research groups underwent this protocol. We replaced “facility” with “colony” for clarity.

Tables 1 and 2 take up several pages and they present information in a level of detail that is more suitable for the Supplementary materials section than for the article proper. Please move these tables to the Supplementary materials, and include only a brief summary of them in the Materials and Methods section.

We have moved the tables to Supplementary Materials. Since the legend of Figure 2 already includes a short description of the training days we did not add another summary of Table 2, but rather extended the figure legend a bit.

Line 270 Latency to approaching a human hand is a relevant parameter to measure in this study, but I don’t think that the main issue here is the stress effect on social interaction. Social interaction is not an appropriate description for how mice react to human handling. Animals interact socially with members of their own species, and may also develop a social relationship to individuals of other species (e.g. dogs, cats, horses, farm animals, rats) but that level of relationship is rarely established with a lab mouse. The human hand is the source of stress and discomfort in handling, and the mice are likely to associate the hand with the negative experience of being handled. Mice who have experienced less stressful handling are likely to show a shorter latency to approach the hand (do the mice really interact with the hand?). Please consider this also elsewhere in the manuscript where this concept is addressed.

We removed the first sentence of this paragraph and ensured that latency to interact is not discussed as a measure of social interaction throughout the manuscript.

Line 333 Please avoid starting a sentence with a digit.

The sentence was rephrased.

Lines 416-417 What does “No remarkable changes were detectable depending on training success or age” mean? Please reword.

We have reworded this sentence.

Lines 433-434 “Cage means (average body temperature of all mice in one cage) were treated as the experimental unit.”. I think most readers will understand what you mean, but to use precise and correct terminology, the cage is the experimental unit, whereas the average body temperature of all mice in a cage is the datapoint.

This was rephrased (also in other figure legends where this applied).

Lines 512-518 How can you measure social interaction when the mouse is alone in the cage with no other mice present?

We measured the latency to interact, this was phrased incorrectly in the figure title and has been adjusted.

Line 566 “variation”, not “variety”

This has been changed.

Line 577 “increasing”, not “maximizing”

This has been changed.

Lines 578-580 Please reword. I suggest “many studies show improved welfare when replacing tail handling by tunnel or cup handling [15, 29] and this beneficial effect remain even when animals are subject to frequent interventions like ip injections.

We reworded the sentence following your suggestion.

Lines 616-617 I have never seen the word falsify being used in this way. In science, the word falsify stands for either the formal testing of hypothesis, or for researcher manipulation of data. Please reword. Also, you refer to “by various circumstances” but then only give one example.

• We have reworded this part.

---

## [Editor Report · Decision Letter 2]

22 Apr 2026

A novel application tunnel in combination with medical training reduces stress induced by frequent intraperitoneal injections and blood draws in mice

PONE-D-26-00796R2

Dear Dr. Hammerich,

We’re pleased to inform you that your manuscript has been judged scientifically suitable for publication and will be formally accepted for publication once it meets all outstanding technical requirements.

Kind regards,

I Anna S Olsson, Ph.D.

Academic Editor

PLOS One
---

## [Editor Report · Acceptance letter]

PONE-D-26-00796R2

PLOS One

Dear Dr. Hammerich,

I'm pleased to inform you that your manuscript has been deemed suitable for publication in PLOS One. Congratulations! Your manuscript is now being handed over to our production team.

Kind regards,

on behalf of

Dr. I Anna S Olsson

Academic Editor

PLOS One